# A global genomic analysis of *Salmonella* Concord reveals lineages with high antimicrobial resistance in Ethiopia

Antimicrobial resistant *Salmonella enterica* serovar Concord (*S*. Concord) is known to cause severe gastrointestinal and bloodstream infections in patients from Ethiopia and Ethiopian adoptees, and occasional records exist of *S*. Concord linked to other countries. The evolution and geographical distribution of *S*. Concord remained unclear. Here, we provide a genomic overview of the population structure and antimicrobial resistance (AMR) of *S*. Concord by analysing genomes from 284 historical and contemporary isolates obtained between 1944 and 2022 across the globe. We demonstrate that *S*. Concord is a polyphyletic serovar distributed among three *Salmonella* super-lineages. Super-lineage A is composed of eight *S*. Concord lineages, of which four are associated with multiple countries and low levels of AMR. Other lineages are restricted to Ethiopia and horizontally acquired resistance to most antimicrobials used for treating invasive *Salmonella* infections in low- and middle-income countries. By reconstructing complete genomes for 10 representative strains, we demonstrate the presence of AMR markers integrated in structurally diverse IncHI2 and IncA/C2 plasmids, and/or the chromosome. Molecular surveillance of pathogens such as *S*. Concord supports the understanding of AMR and the multi-sector response to the global AMR threat. This study provides a comprehensive baseline data set essential for future molecular surveillance.

*S*almonella is a diverse genus comprising over 2500 serovars which can cause a range of infections[1]. Typhoidal *Salmonella* serovars Typhi and Paratyphi A exclusively infect humans and cause systemic disease[2]. Infections with non-typhoidal *Salmonella* (NTS) exemplify the interconnection between humans, animals, and the environment, captured in the "one health" concept[3]. Many NTS serovars have a zoonotic reservoir, are primarily associated with foodborne transmission and can persist for extended periods in the environment[2,4]. In humans in general, NTS mainly cause gastroenteritis, and in approximately 5% of the cases bloodstream infection occurs due to underlying conditions[5,6]. In sub-Saharan Africa, however, NTS more often causes life-threatening invasive NTS (iNTS) infections, particularly in HIV-positive individuals or young children with *Plasmodium falciparum*

malaria, anaemia and/or malnutrition[7–10]. Due to the high fatality rates, these invasive infections require prompt antimicrobial treatment[11].

*Salmonella enterica* serovar Concord (hereafter *S*. Concord) is an infrequently reported NTS serovar. The first *S*. Concord isolates were described in 1944, originated from the USA and UK, and were isolated from poultry and human stool respectively[12]. Four other reports demonstrated the presence of *S*. Concord in Ethiopia between 1974 and 1981[13,14] and in Saudi Arabia in 1982[15,16]. A sudden increase in human infections caused by *S*. Concord was noted from 2003 onwards in Europe and the USA[17–24]. This dissemination of *S*. Concord was linked to the international adoption of Ethiopian children[20,24]. Adopted children testing positive for *S*. Concord were either asymptomatic or showed mild abdominal discomfort[20,25], had diarrhoea or bloody diarrhoea,

e-mail: wim.cuypers@uantwerpen.be; sandra.vanpuyvelde@uantwerpen.be

occasionally with fever[24], and a study from Ethiopia demonstrated that 30.6% of *S*. Concord infections were bloodstream infections[26]. After 2012, no additional reports on *S*. Concord were published until 2018, when foodborne *S*. Concord outbreaks were reported in Israel and the USA[27,28].

The dissemination of *S*. Concord from Ethiopia raised public health concerns due to the frequent occurrence of multidrug resistance (MDR), defined for *Salmonella* as co-resistance to ampicillin, chloramphenicol and trimethoprim/sulfamethoxazole[18–23,26]. Recommended antibiotic treatment of iNTS infections with MDR strains are ceftriaxone, a third-generation cephalosporin and ciprofloxacin, a fluoroquinolone[29,30]. Alarmingly, MDR *S*. Concord isolates often produced an extended-spectrum beta-lactamase (ESBL) conferring ceftriaxone resistance[20]. Less frequently, decreased ciprofloxacin susceptibility (DCS) has been described in addition to MDR and ceftriaxone resistance[18–23,26], resulting in extensive drug resistance (XDR) phenotypes[31]. The macrolide azithromycin is a promising candidate for oral treatment of XDR *Salmonella* infections in low- and middle-income countries (LMICs)[31] and prior to this study azithromycin resistance was not reported in *S*. Concord. The carbapenem antimicrobial meropenem can be used as last-resort treatment option, but its availability in LMICs may be limited. Therefore, co-resistance to all recommended, accessible and affordable antibiotics, *i.e.* MDR combined with ceftriaxone resistance, DCS and azithromycin resistance, is referred to as pandrug resistance (PDR) for iNTS in LMIC settings[31].

In this work, we study 284 predominantly human *S*. Concord isolates collected over 78 years to reconstruct the population diversity, evolution, and antimicrobial resistance (AMR) distribution of *S*. Concord. We show that *S*. Concord is a diverse and polyphyletic serovar and identify several lineages of which four circulated in Ethiopia and were mainly MDR and ceftriaxone-resistant. Other lineages show a more global spread, lack AMR markers, and harbour isolates linked to food sources in addition to human isolates.

## Results

### Concord is a polyphyletic *Salmonella* serovar found in three diverse super-lineages

High-quality whole-genome sequence data was used to reconstruct the population structure of the 284 confirmed *S*. Concord isolates by inferring a neighbour-joining tree based on core-genome multilocus sequence typing (cgMLST) distances. Three polyphyletic *Salmonella* super-lineages, i.e. genetically diverse groups with up to 2000 allelic differences[32], were identified, namely HC2000_750, HC2000_141 and HC2000_177997, and are hereafter referred to as super-lineage A, B and C respectively (Fig. 1A). Querying the EnteroBase database for isolates that were part of these three super-lineages resulted in identifying a set of 1430 isolates comprising eight different serovars, all with the same O:7 serogroup (formerly known as group C1) (Fig. 1B).

Super-lineage B was the largest and most diverse super-lineage consisting of 2.8% (28/981) of *S*. Concord isolates analysed here, as well as seven additional serovars including Mikawasima (*n* = 667), Potsdam (*n* = 145), Irumu (*n* = 63), Bonn (*n* = 41), Richmond (*n* = 32), Amersfoort (*n* = 4) and Hartford (*n* = 1) (Fig. 1B). A maximum likelihood phylogeny of *S*. Concord super-lineage B isolates showed that one isolate from a patient who reported travel to Madagascar was distantly related to the remaining isolates of which two originated from England and Belize (Supplementary Fig. 1).

Super-lineage C harboured exclusively 14 *S*. Concord isolates, of which 12 (10 human and two environmental) were sampled during a foodborne outbreak in the USA in 2019. The source of the outbreak was tracked to Tahini products imported from Israel[33]. Co-clustering of two isolates collected in Czechia with the remaining isolates indicates a broad geographical spread of this super-lineage (Supplementary Fig. 2).

Most *S*. Concord isolates (*n* = 245) were part of super-lineage A, which consisted of 98.8% (245/248) of *S*. Concord and three isolates of

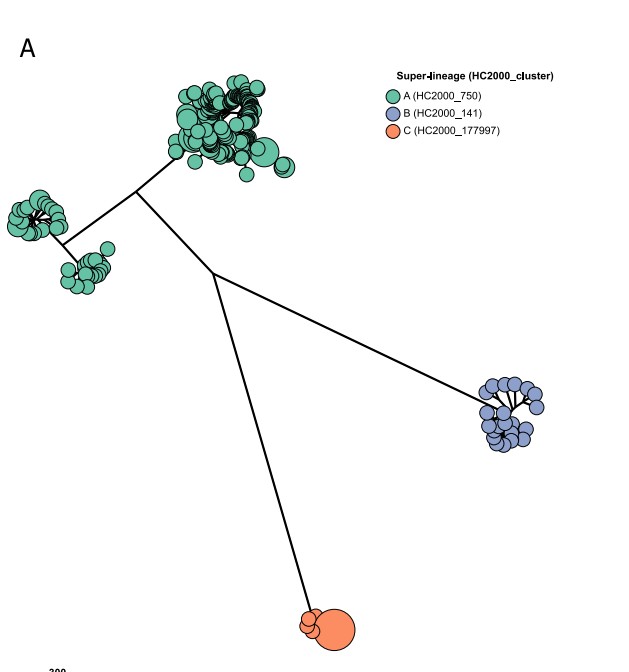

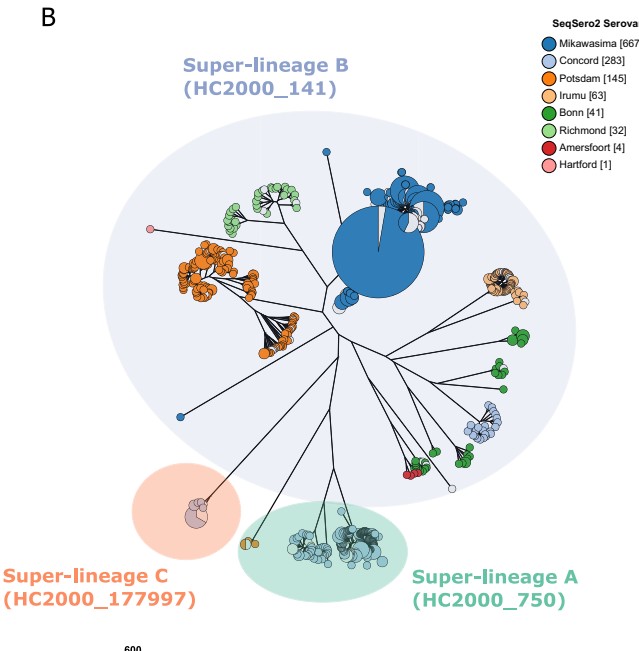

**Fig. 1 | Three polyphyletic super-lineages of *S*. Concord. A** Neighbour-joining cgMLST tree constructed using EnteroBase showing three different super-lineages in the *S*. Concord collection. Dots are coloured according to the super-lineage. **B** Overview of all *Salmonella* isolates present in EnteroBase on 21/07/2022 that cluster together with *S*. Concord in the super-lineages HC2000_750, HC2000_141 and HC2000_177997. Dots in the tree are coloured according to the serovar determined by SeqSero2. The data required to reproduce these figures is available via EnteroBase. Scale bars represent the number of cgMLST allele differences.

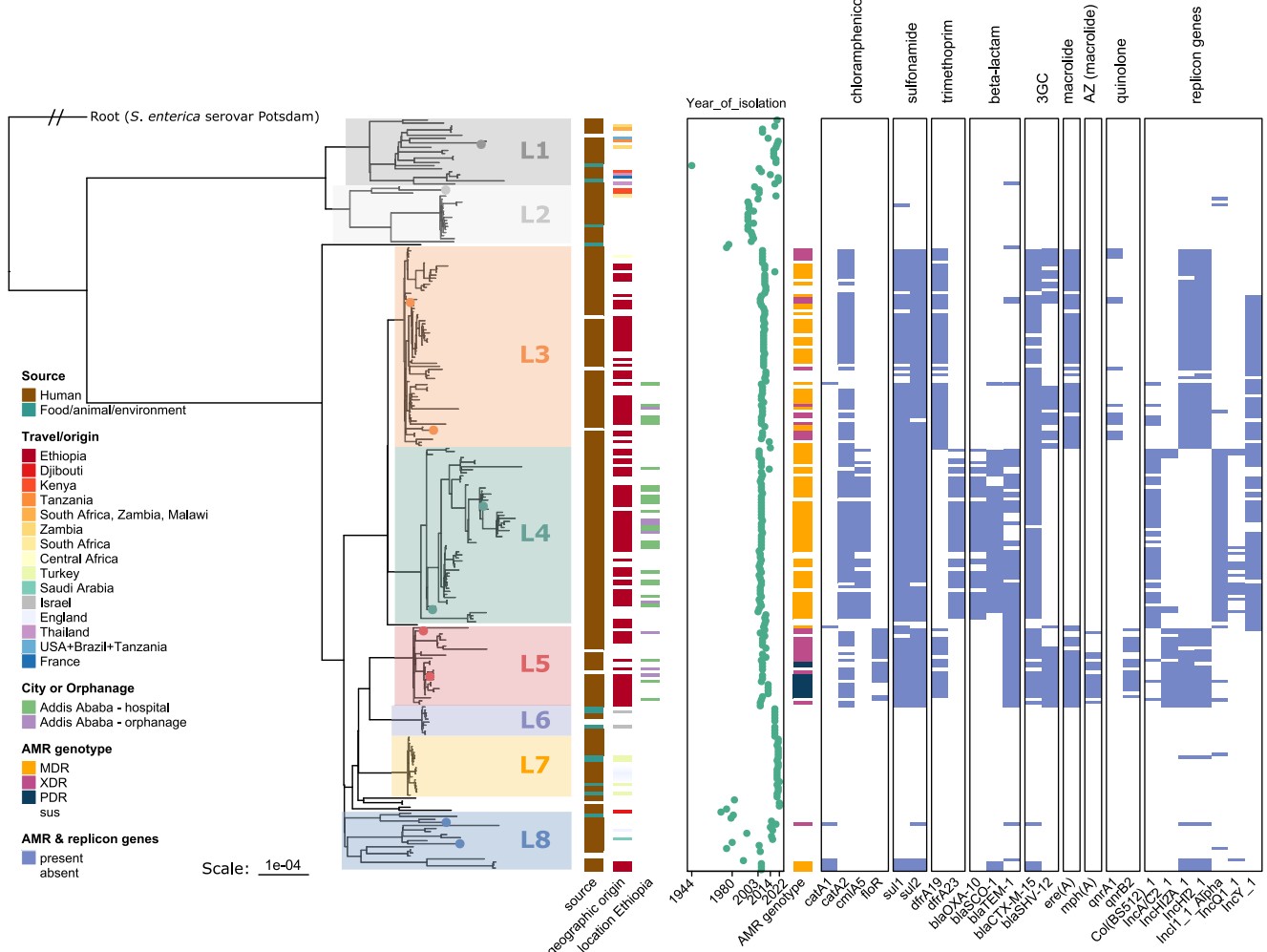

**Fig. 2 | Overview of the population structure, metadata, antimicrobial resistance genes and plasmid replicon types of *S.* Concord super-lineage A.** Maximum likelihood phylogeny for 245 *S.* Concord isolates belonging to super-lineage A and based on 7148 non-recombinant core SNPs. The tree was rooted on a closely related *S. enterica* serovar Potsdam isolate that was also part of super-lineage A. Lineage names are indicated in each coloured square (e.g. Lineage 1 = L1). Tips highlighted with a coloured dot were selected for long-read sequencing. Ggtree v2.2.4 was used to plot the tree. Metadata is shown at the right side of the tree together with a summary of commonly occurring AMR genotypes and the presence- (blue cells) or absence (white cells) of common AMR genes and replicon types. The scale bar indicates the number of substitutions per site. Complex-Heatmap v2.8.0 was used to for plotting. Abbreviations: AMR = antimicrobial resistance, MDR = multidrug resistance, ESBL = extended-spectrum beta-lacta-mase, XDR = extensive drug resistance, PDR = pandrug resistance, sus = suscep-tible, 3GC = third-generation cephalosporin, AZ = azithromycin.

serovar Potsdam (Fig. 1B). Previously studied isolates linked to Ethiopia and presenting high antimicrobial resistance[20,21,26] were all part of super-lineage A.

## Circulation of multiple *S.* Concord super-lineage A lineages in Ethiopia

Phylogenetic analysis of 245 *S.* Concord isolates within super-lineage A and subsequent cluster identification revealed the presence of 8 monophyletic lineages which we named Lineage 1 (L1) up to lineage 8 (L8) (Fig. 2).

L1 and L2 were a sister group of all *S.* Concord lineages and had no confirmed link with Ethiopia (Fig. 2; Supplementary Fig. 3). Their geographic origin was diverse, but eight isolates were linked to countries in South- and East Africa (Supplementary Fig. 3). One of the first of four *S.* Concord isolates ever described, isolated in 1944 in the USA from poultry[12], was part of L1.

Four lineages, L3, L4, L5 and L8 (*n* = 161) exclusively harboured human isolates including all isolates with a confirmed epidemiological link to Ethiopia (*n* = 98), four isolates linked to other regions or countries (England, the Central Africa region, and Saudi Arabia), and 51 isolates for which no geographical information was available (Fig. 2). The majority of isolates linked to Ethiopia were collected outside Ethiopia (*n* = 73). Isolates from outpatients in Ethiopia (*n* = 25) and isolates from adoptees from specific orphanages in Addis Ababa (*n* = 8) were closely related and co-occurred within the three largest lineages L3, L4 and L5 (Fig. 2). The years of isolation within these lineages ranged from 2003 to 2018 (Fig. 2). Hence during the beginning of this century, three *S.* Concord lineages simultaneously circulated in Ethiopian orphanages and the community. One of these lineages, L4, exhibited a greater invasiveness index predicted from genomic features (Supplementary Fig. 4), likely due to gene presence-absence variations in genes previously linked to increased invasiveness (Supplementary Table 1).

Recent Ethiopian isolates were evolutionary closely related to historical isolates. Lineage 8 (*n* = 23) harboured eight historical isolates (isolated between 1975 and 1993) without known geographic origin, three recent isolates from Ethiopia isolated in 2006, and 8 recent isolates without known geographic origin (Supplementary Fig. 5).

**Table 1 | Distribution of antimicrobial resistance (AMR) genes across the chromosome and different types of plasmids found in representative super-lineage A S. Concord genomes**

| Lineage | Lab ID | Location | Size (bp) | AMR genes | Insertion sequences or transposons in AMR cassettes |
|---|---|---|---|---|---|
| 3 | 95907 | Chromosome | 14495 | *sul2, aph(3″)-Ib, aph(6)-Id* | IS26, IS5075, ISStma11 |
| | | IncHI2 plasmid | 288445 | *catA2, dfrA19, sul1, sul2, bla*$_{CTX-M-15}$, *bla*$_{SHV-12}$, *qnrA1* | IS1353, IS26, Tn2, ISEc63, IS1247, IS6100, ISPa38, IS5075, IS903 |
| 3 | 64206 | Chromosome | 13168 | *sul2, aph(3″)-Ib, aph(6)-Id* | IS26, ISStma11 |
| | | IncHI2 plasmid | 372354 | *catA2, dfrA19, sul1, sul2, bla*$_{CTX-M-15}$ | IS26, Tn2, ISEc63, IS1247, IS6100, ISPa38, IS5075 |
| 4 | 1035531 | Chromosome | 91371 | *bla*$_{SCO-1}$, *bla*$_{OXA-10}$, *dfrA23, catA2, cmlA5, sul1, sul2, bla*$_{CTX-M-15}$ | IS1353, IS26, IS4321R, ISCfr1, ISPa40, ISPmi2, ISStma11, TnAs1 |
| | | IncI1α plasmid | 85904 | *blaTEM-1* | Tn3, Tn2 |
| 4 | 254833 | Chromosome | 55792 | *bla*$_{OXA-10}$, *dfrA23, catA2, cmlA5, sul1, sul2* | IS1353, IS26, IS4321R, ISCfr1, ISPa40, ISStma11, TnAs2 |
| | | IncA/C2 plasmid | 71024 | *dfrA7, sul1, bla*$_{CTX-M-15}$ | IS26, ISEcp1 |
| | | IncI1α plasmid | 85904 | *bla*$_{TEM-1}$ | Tn3, Tn2 |
| 5 | 0508H45184 | IncHI2/IncA/C2 hybrid plasmid | 415479 | *dfrA12, sul1, sul2, floR, bla*$_{TEM-1}$, *catA2, bla*$_{CTX-M-15}$, *mph(A)* | Tn2, ISEc63, IS26, ISEcp1, IS103, ISKpn8, ISEsa1, ISEsa2, IS1D, IS1R, ISBcen27, ISKpn21, IS903, ISVvu2, Tn3 |
| 5 | 70366 | Chromosomally integrated IncHI2 plasmid | 270994 | *dfrA19, sul2, catA2, bla*$_{TEM-1}$, *bla*$_{SHV-12}$, *qnrB2* | IS103, IS26, IS4321R, IS903, ISBcen27, ISEc63, ISEsa1, ISEsa2, ISKpn21, ISKpn8, Tn2 |
| | | IncA/C2 plasmid | 160357 | *floR, bla*$_{TEM-1}$, *sul1, sul2, mph(A), bla*$_{CTX-M-15}$ | IS26, IS1247, IS6100, ISEcp1, Tn2 |
| 8 | 850890 | IncHI2 plasmid | 294257 | *bla*$_{TEM-1}$, *dfrA14, sul2, catA1, bla*$_{OXA-1}$, *bla*$_{CTX-M-15}$, *qnrB1, aac(6′)-Ib-cr5* | IS26, IS1×2, IS6100, IS3000, ISKpn11, Tn2, ISPa38, IS5075, ISEcp1, IS1R, IS1D |

Reference genomes were constructed from Oxford Nanopore reads and polished with Illumina reads. The size column reports the size in base pairs (bp) of each chromosomal integration or plasmid. Only AMR genes that were exact database matches and contributed to the MDR, XDR or PDR genotype were listed. Reference genomes for three susceptible isolates (with laboratory identifiers 679052, H044240362 and H04340470) were not shown.

## Transmission from food to humans in super-lineage A lineages
Starting from 2019, the CDC and FDA monitored a foodborne outbreak in the USA caused by Tahini and Halva food products manufactured in Israel[34]. Isolates originating from these food products were part of L6 and L7 and showed little genetic variation compared to other lineages, with maximum seven and nine pairwise SNP differences for L6 and L7 respectively (Supplementary Figs. 6–8).

Nine out of 10 L6 isolates were isolated in 2018. Likely giving rise to an outbreak, L6 spread beyond the USA as it contained a mixed occurrence of isolates from both tahini from Israel as well as human stool isolates that were isolated in France, the UK, and the USA. One stool isolate obtained in France differed by one core SNP from an isolate recovered from tahini in Israel (Supplementary Fig. 7).

Within L7, four food isolates from food products (halva, nut spread) isolated in Turkey (n = 4) were closely related to human isolates from the UK and France (n = 16). The isolates were highly similar with one up to nine pairwise core SNP differences, while the year of isolation ranged from 2020 to 2022 (Supplementary Figure 8). Similarly, one human isolate differed by one SNP from a halva isolate despite being isolated in England and Turkey, respectively (Supplementary Fig. 8).

## Occurrence of MDR, XDR and PDR in S. Concord lineages from Ethiopia
We identified 50 AMR markers in total, including 49 resistance genes and one AMR-conferring SNP (Supplementary Figure 9). Clinically relevant markers were further analysed.

MDR + ESBL, XDR or PDR markers were found in almost half (47%) of the S. Concord super-lineage A isolates and were associated with L3, L4, L5 and L8 that harboured isolates from Ethiopia predominantly collected between 2003 and 2014 (Fig. 2). MDR and ESBL genes, associated with co-resistance to ampicillin, chloramphenicol, trimethoprim/sulfamethoxazole and ceftriaxone, were first detected in the population from 2003 onwards and co-occurred in 34.3% (84/245) of the super-lineage A isolates. Genotypic XDR and PDR, implying additional DCS and/or azithromycin resistance, occurred from 2005 onwards in 11% (27/245) and 3.7% (9/245) of the super-lineage A isolates respectively and predominantly within L2 and L5 (Supplementary Fig. 10).

A decrease in highly resistant isolates was observed after 2014 as two out of 82 isolates contained MDR + ESBL genes. The geographic origin of these isolates was undetermined, and they clustered in L3 and L8 together with isolates from Ethiopia.

## Multiple introductions and rearrangements of AMR genes
A fine-grained view on AMR and plasmid replicon types indicated that AMR was introduced multiple times resulting in inter-lineage variation of the S. Concord super-lineage A accessory genome (Fig. 2). Out of 17 genes that contributed to MDR, PDR and XDR, three occurred at least once in all four highly resistant lineages, i.e., sulphonamide resistance genes *sul1* and *sul2*, and the ESBL gene *bla*$_{CTX-M-15}$ (Fig. 2). The other 14 genes, showing combinational differences between lineages, encoded resistance to six categories of antimicrobials. For example, genes linked to DCS, i.e., *qnrA1* (n = 14) were found in L3 and *qnrB2* (n = 11) in L5, whereas the azithromycin resistance gene *mph(A)* (n = 10) was uniquely found in L5 (Fig. 2). Plasmid replicon types present in at least five isolates included Col(BS512), IncA/C2, IncHI2A, IncI1α, IncQ1, IncY (Fig. 2).

Both the chromosome and plasmids can carry AMR determinants in S. Concord[20]. To assess structural rearrangement, we reconstructed ten complete genomes from isolates representing different AMR profiles and lineages and compared the variation between regions containing AMR genes. Complete genomes of resistant isolates all displayed a unique partitioning of AMR genes between the chromosomes and plasmids (Table 1).

Large chromosomal AMR regions were flanked by the IS1R element in five completed genomes originating from L3, L4 and L5 (Supplementary Fig. 11). The largest region flanked by IS1R was found in L5 and corresponded to a large (271 kb) IncHI2 plasmid conferring XDR (Table 1). Smaller chromosomal integrations in closely related L3 and L4 isolates had the same start-and-end sequence, but L4 acquired additional AMR genes flanked by different types of insertion sequences such as IS26 (Table 1; Supplementary Fig. 12).

Plasmids displayed unique structural arrangements within and between lineages (Table 1). IncHI2 plasmids, occurring in L3, L8, and chromosomally integrated in L5, had a conserved backbone, and differed in those regions harbouring AMR markers (Supplementary Figure 13). Each AMR region harboured multiple insertion sequences (Table 1). A 415 kb circular contig of a complete L5 genome contained backbone elements originating from both IncHI2 and IncA/C2 plasmids present in L3 and L5 (Supplementary Fig. 14). It is highly probable that this was an IncHI2A/IncA/C2 hybrid plasmid.

## High concordance between MDR, PDR and XDR genotype and phenotype

A selection of 56 isolates covering multiple *S.* Concord super-lineage A lineages and different clinically relevant genotypic AMR combinations (susceptible, MDR, XDR and PDR), were subjected to antimicrobial susceptibility testing (AST) to assess the association of AMR genotype with its phenotype. Of the 56 isolates tested, 32 showed phenotypic MDR, nine showed XDR and three showed PDR (Supplementary Data 2). Overall, the AMR genotype was a good predictor of phenotypic MDR, XDR and PDR in *S.* Concord with a high concordance between genotype and phenotype, ranging from 98% to 100% (Supplementary Table 2). The contribution of each combination of resistance genes to the minimum inhibitory concentration (MIC) of an antimicrobial compound is shown in Supplementary Fig. 15.

## Alternative antimicrobials with in vitro activity against *S.* Concord

In addition to the recommended antimicrobials to treat (potentially) invasive *Salmonella* infections, we explored the in vitro susceptibility of the same set of isolates as described earlier against several alternative antimicrobials including gatifloxacin, meropenem, temocillin, tigecycline, colistin, and several combination agents containing beta-lactamase inhibitors, namely ceftazidime/avibactam, ceftolozane/tazobactam, meropenem/vaborbactam, piperacillin/tazobactam.

All tested isolates were susceptible to gatifloxacin, ceftazidime/avibactam, meropenem and meropenem/vaborbactam. Resistance to colistin ($n = 1$), tigecycline ($n = 1$), piperacillin/tazobactam ($n = 3$) and ceftolozane/tazobactam ($n = 6$) was rare. Similar as described earlier for *Salmonella* and other Enterobacterales[35,36], the mobilised colistin resistance gene *mcr-9.1* was not associated with phenotypic colistin resistance (Supplementary Fig. 16). *mcr-9.1* was present in 15 isolates selected for AST including one resistant isolate (MIC = 8 μg/mL; isolate 64206 in Supplementary Data 2). The colistin-resistant isolate had a disrupted *mgrB* gene due to insertion of IS5 family transposase IS903. *mgrB* disruption has been described to be causative for colistin resistance before in other Enterobacterales[37,38]. Temocillin is an antimicrobial to treat infections with beta-lactamase-producing bacteria[39], but 35 out of 56 tested isolates *S.* Concord isolates were resistant, likely due to the presence of $bla_{\text{CTX-M-15}}$ or combinations of multiple beta-lactamase genes (Supplementary Fig. 16).

## Discussion

Between 2001 and 2009, antimicrobial-resistant *S.* Concord was found across the globe among adoptees from Ethiopia[20,24,25] and children in Ethiopia[26]. Additionally, there were occasional reports of *S.* Concord outside of Ethiopia, but it remained unclear how *S.* Concord isolates were linked to each other. Here, we have compiled the largest collection of *S.* Concord genomes that are internationally available to date, comprising 284 isolates from 12 countries and spanning 78 years. We unveiled the population structure of *S.* Concord and found that *S.* Concord is a diverse polyphyletic serovar spread across three *Salmonella* super-lineages. Resultantly, this advocates for the use of whole-genome sequencing analyses for surveillance and public health investigations. The data generated here and made publicly available provides a framework to place further work on *S.* Concord in context.

Super-lineage A was further subdivided into eight *S.* Concord lineages. Antimicrobial resistance was almost exclusively found in four *S.* Concord lineages with isolates linked to Ethiopia and historical isolates of unknown geographic origin. Earlier works report *S.* Concord in Ethiopia since the 1970s[13,14,26]. It is likely that these lineages have been endemic in Ethiopia for decades. High antimicrobial use subsequently contributed to the natural selection of more resistant strains in Ethiopia, which spread due to the limited access to good hygiene and sanitation practices[20,24]. Our genomic analysis showed that isolates related to orphanages circulated in the community in Addis Ababa corroborating the earlier findings of Beyene et al. (2011)[26]. Hence, highly resistant *S.* Concord identified in the study was not limited to the orphanage environment in Ethiopia, but instead may have been identified at higher rates by reference laboratories in Europe and the USA via adoptees originating from Ethiopian orphanages, which introduced a sampling bias. Due to missing metadata, we could not estimate the level of transmission between the community and the orphanages, as well as the role of different actors. Sample bias and missing metadata are two major limitations to this study, which are most often intrinsic to studies relying on historical data from infections linked to low-resource settings and sentinel surveillance data.

Accessory genomes showed substantial genetic variation shaped by multiple AMR gene acquisition events associated with IncHI2, IncA/C2, and IncI plasmids, chromosomal integration facilitated by the IS1R element, and additional rearrangements via IS26 and/or other insertion sequences. IncHI2 plasmids carrying genes encoding XDR and PDR are responsible for high AMR in the ongoing invasive *S. enterica* serovar Typhimurium (*S.* Typhimurium) epidemic in SSA[7,40,41]. IS1R is a known mediator of chromosomal integration of AMR in Enterobacterales[42,43] and IS26 mediated structural differences in the AMR regions of *Salmonella* genomic island 1 in *S. enterica* serovar Kentucky[44]. Chromosomal integration of AMR, which we observed in L3, L4, L5 and L8, was reported previously in *S.* Concord[20] and other salmonellae including *S.* Typhimurium DT104[45,46] and other Enterobacterales[47,48]. Chromosomal integrations are considered an effective mechanism for the stable dissemination of AMR genes[47]. However, in our dataset, there is no indication that the chromosomal integration of AMR has facilitated the expansion of resistant *S.* Concord since both L3 (plasmid-encoded AMR in two reference genomes) and L4 (chromosomally located AMR in two reference genomes) harboured more isolates than other lineages and were sampled in the same period.

The development of MDR + ESBL, XDR and PDR in *S.* Concord through horizontal gene transfer illustrates the alarming AMR situation in LMICs. In 2011, Beyene et al.[26] predicted the potential spread of highly resistant *S.* Concord from Ethiopia. In isolates collected after 2014, AMR was rare. This could be due to sample bias, since no recent reports on *S.* Concord in Ethiopia were available. Alternatively, recent susceptible strains may have emerged from an unknown reservoir. Between 2016 and 2020, MDR and ESBL production was still highly abundant in other Enterobacterales isolated from patients in Addis Ababa[49–51], implying that AMR itself is circulating in the region. Systematic community-based and hospital-based surveillance is required to elucidate the current burden of resistant *S.* Concord in Ethiopia, and to determine whether PDR *S.* Concord with very limited available treatment options persists. We found that PDR isolates were susceptible to the antibiotics gatifloxacin, ceftazidime/avibactam and meropenem but to date many of these antimicrobials are not well established as treatment alternatives for invasive *Salmonella* infections due to (i) limited availability of data from clinical and pharmacokinetic studies that allow the formulation of evidence-based treatment recommendations and (ii) limited availability and affordability in LMICs.

The reservoir of resistant *S.* Concord remains unknown. IncHI2 and IncA/C2 type plasmids described in this study have been

previously mentioned to possibly have an animal origin[52], and *S.* Concord was previously isolated from animals[15,16]. Similar to iNTS in Kenya[53], asymptomatic carriage and human-to-human transmission has, however, been described[25], which could indicate a role of humans as a reservoir for resistant *S.* Concord from Ethiopia. Transmission of *S.* Concord from food to humans was supported by data from L6 and L7 which likely represented two foodborne outbreaks linked to sesame seed-containing food products from Israel and Turkey. The FDA and CDC previously reported these outbreaks[28], and we identified that the outbreak strain spread beyond the USA to the UK and France. There was a close phylogenetic relationship with isolates from Ethiopia, but no direct link since we mainly had access to clinical isolates. As both Israel and Turkey import sesame seeds from Ethiopia[54] it is possible that there is a geographical link with Ethiopia. A more broad sampling campaign is required to further unravel *S.* Concord transmission routes.

In conclusion, we described how the highly resistant *S.* Concord circulating in Ethiopia is positioned within the diverse and polyphyletic *S.* Concord population. Isolates linked to Ethiopia from L3, L4, L5 and L8 accumulated AMR genes responsible for MDR, XDR, and PDR encoded on diverse plasmids or on the chromosome. The example of *S.* Concord illustrates how AMR can disseminate globally through human travel. Susceptible *S.* Concord strains appear to have disseminated recently through the food chain. Molecular surveillance is critical to monitor the spread of resistant foodborne pathogens, and is expected to underpin updates of treatment protocols and further support development of prevention policies. This requires a comprehensive framework obtained through genomics studies of historical and contemporary strains to unravel the microbiological and epidemiological events underlying the emergence and dissemination of highly resistant bacterial strains. For *S.* Concord, such a framework was missing, despite previous reports of alarmingly high levels of AMR[18–20,22–26,55]. Greater focus and urgency is required in a global effort to contain the alarming spread of AMR

## Methods

### Ethics statement

This study relied exclusively on bacterial isolates and associated metadata collected under local mandates for laboratory-based surveillance of bacterial pathogens and organisms. In the event that the listed institutes have collected personal identifiable information, it was deleted during the preliminary phase of data collection for this project. Therefore, in all instances, neither informed consent nor approval from an ethics committee was required. Isolates or genome sequences were obtained from the following institutes: Institut Pasteur (France), Technical University of Denmark (DTU Denmark, Denmark), the United Kingdom Health Security Agency (UKHSA, United Kingdom), Jimma University (Ethiopia), US Centres for Disease Control and Prevention (USA), Institute of Tropical Medicine Antwerp (Belgium), Sciensano (Belgium), The Center for Food Safety and Applied Nutrition (CFSAN) of the Food and Drug Administration (FDA) (USA), and the National Institute of Public Health Czechia (Czechia). All collection efforts were in line with local laws and regulations. MTAs were appropriately established for the isolates shipped to ITM Antwerp from DTU Denmark, UKHSA and Sciensano. Each partner ensured that all proper national rules and regulations were followed throughout the project.

### *S.* Concord sequence data collection

Short-read sequencing data of 310 *S. Concord* isolates were initially included in the study, of which 126 originated from the public domain and 184 were generated as part of this study. Metadata was collected in Microsoft Excel v16.0, listed in Supplementary Data 1, and included in Enterobase (https://enterobase.warwick.ac.uk/species/senterica/ search_strains?query=workspace:79416). In total, 26 isolates were removed from further analysis because they failed sequencing quality control ($n = 8$), were not confirmed as serovar Concord by SeqSero2 ($n = 15$) or could not be linked to an original laboratory identifier ($n = 2$). One isolate was removed because only patient-unique isolates were considered.

The resulting set of 284 isolates originated from different isolate collections and included 117 isolates from Institut Pasteur (Paris, France), 46 from the Technical University of Denmark (DTU-Food) (Kgs. Lyngby, Denmark), 44 from the United Kingdom Health Security Agency (UKHSA) (London, UK), 26 from Jimma University (Jimma, Ethiopia), 25 from the US Centres for Disease Control and Prevention (CDC) (Atlanta, USA), nine from the Institute of Tropical Medicine (ITM) Antwerp (Antwerp, Belgium), eight from Sciensano (Elsene, Belgium), seven from The Center for Food Safety and Applied Nutrition (CFSAN) of the Food and Drug Administration (FDA) (USA), and two from the National Institute of Public Health (NIPH) (Czechia).

The isolates were recovered in 12 different countries, including France ($n = 116$), UK ($n = 53$), USA ($n = 39$), Belgium ($n = 17$), Ethiopia ($n = 27$), Austria ($n = 9$), The Netherlands ($n = 4$), Denmark ($n = 4$), Turkey ($n = 4$), Czechia ($n = 2$), Ireland ($n = 2$), and Israel ($n = 2$). For five isolates the country of isolation was unknown.

The great majority of the isolates were from humans (90.5%, 257/284). For 63.8% (164/257) of the human isolates the specimen types were known and included stool ($n = 148$), blood ($n = 10$), urine ($n = 3$), pus ($n = 2$), and meconium ($n = 1$). The 5.3% (15/284) non-human isolates originated from halva ($n = 3$), tahini ($n = 2$), poultry ($n = 1$), and nut spread ($n = 1$), or the isolation source was not further specified than 'food' ($n = 4$) or 'environment' ($n = 3$). The source was unknown for 4.2% (12/284) of the isolates.

The geographic origin was available for 45.5% (129/284) of the isolates. Most isolates were linked to Ethiopia ($n = 101$). These isolates originated from Ethiopian adoptees, were isolated from patients in Addis Ababa, or were linked to travel to Ethiopia. For 28 other isolates the travel history of a patient or the exact origin of the isolate was known: UK ($n = 4$), Turkey ($n = 4$), Kenya ($n = 3$), Zambia ($n = 2$), Thailand ($n = 2$), Israel ($n = 2$), Belize ($n = 1$), Central Africa ($n = 1$), Djibouti ($n = 1$), France ($n = 1$), Madagascar ($n = 1$), Saudi Arabia ($n = 1$), South Africa ($n = 1$), Tanzania ($n = 1$), USA ($n = 1$). One patient reported travel to the USA, Brazil and Tanzania, and one patient reported travel to South Africa, Zambia and Malawi ($n = 1$).

The year of isolation was available for 95.8% (272/284) isolates and ranged from 1944 to 2022. Fifty-percent of the isolates had been collected between 2006 and 2017. Throughout this work, we refer to isolates from the historical collection of Institut Pasteur, isolated between 1970 and 2000, as 'historical isolates', and isolates collected from 2000 onward as 'recent isolates'.

### Reference genome assembly

To obtain a representative, high-quality reference genome for short-read mapping, *S.* Concord isolate ITM_8091960 was sequenced using the PacBio RSII system (Pacific Biosciences, California, USA). This isolate originated from a stool sample of an Ethiopian child that presented at the travel clinic of ITM Antwerp in 2008. DNA was prepared using the PacBio Template Prep Kit (Pacific Biosciences, California, USA) and the BluePippin™ system for size selection, for sequencing with the PacBio RSII system at the Wellcome Sanger Institute (Hinxton, UK). Reads were assembled de novo using the HGAP[56] protocol v3.0 implemented in smrtanalysis v2.3.0, resulting in one chromosome-sized contig, and five small contigs. Circlator[57] v1.5.5 was used to remove self-compatible ends and rearrange the start positions of the contigs at the *dnaA* gene, or a predicted gene. The resulting contigs were polished twice using Quiver[56].

## Short-read sequencing and quality control

Illumina (San Diego, California, USA) short-read sequencing data was obtained for 184 isolates sequenced for this study. In brief, all isolates were cultivated on tryptic soy agar (TSA) plates or BD Columbia Agar with 5% sheep blood (Becton Dickinson GmbH, Heidelberg, Germany) and incubated overnight at 37 °C. To ensure purity, a single colony was sub-cultured and again incubated overnight. Next, a single colony or swipe was added to 10 mL Tryptic soy broth (TSB) or Lysogeny broth (LB) for overnight incubation with agitation. From this liquid culture, 300 μL was added to a 1.5 mL Eppendorf tube for DNA extraction with the Gentra Puregene Yeast/Bact. Kit (Qiagen, Hilden, Germany) according to the manufacturer's protocol. Sequencing was performed at the Wellcome Sanger Institute (Hinxton, UK) on the Illumina HiSeq X10.

For Illumina data of the 126 isolates that were sequenced by other institutions, isolates from UKHSA were sequenced as described in Chattaway et al. 2019[58], isolates from Jimma University were sequenced as described in Perez-Sepulveda et al. 2021[59], seven isolates from Institut Pasteur (lab ids: 156 K, 202109373, 202101195, 201804751, 202205355, 202001598, 202005029) were sequenced as described in Jones et al. (2019)[60] and isolates from the CDC and FDA collections were sequenced as part of the PulseNet surveillance system[61].

All short-read data were made publicly available. Accession numbers are listed in Supplementary Data 1.

Raw sequencing reads were trimmed with Trimmomatic[62] v0.39. Read quality was assessed with FastQC (https://www.bioinformatics.babraham.ac.uk/projects/fastqc/) v0.11.9 and MultiQC[63] v1.8. Sequencing quality was further assessed in terms of estimated read depth and mapping-based coverage. The estimated read depth was calculated as the number of sequenced bases divided by the length of the *S.* Concord reference genome. Coverage was calculated as the fraction of the reference chromosome that was covered by at least 10 reads with base quality 20 and mapping quality 60. For this purpose, reads were aligned to the reference genome using minimap2[64] v2.17-r941, clipped alignments were filtered out using samclip (https://github.com/tseemann/samclip) v0.3.0, and bam files were further processed using samtools[65] v1.9. Nine isolates in total with estimated read depth <40 and coverage <85% were discarded. Taxonomic read classification with Kraken2[66] v2.0.8-beta confirmed that sequencing reads originated from the genus *Salmonella* and not a contaminant. Reads were assembled using Spades[67] v3.15.5 with the option '--careful'. Draft assembly quality metrics were generated with Quast[68] v5.0.2.

## Annotation and AMR determinant detection

Reference genomes and draft assemblies were annotated with Prokka[69] v1.14.6 and subjected to in silico serotype determination using SeqSero[70] v1.2.1. In addition, all genome sequences were compared to the PlasmidFinder[71] database (retrieved on March 1, 2020) with Abricate (https://github.com/tseemann/abricate) v0.9.9 to type replicon genes. Only hits with 90% identity and 90% coverage were retained. The presence of known antimicrobial resistance genes and chromosomal point mutations was assessed using AMRFinder[72] v3.10.30 with the 'organism' option set to 'Salmonella' (database version 2022-05-26.1).

## Phylogenetic analyses

A neighbour-joining tree of the entire set of *S.* Concord isolates was constructed based on core-genome multilocus sequence typing (cgMLST) distances. This analysis was caried out using Enterobase platform and relied on the cgMLST V2 plus HierCC V1 scheme, the NINJA algorithm and the GrapeTree[73] visualisation tool. The resulting tree is publicly available via https://enterobase.warwick.ac.uk/ms_tree?tree_id=81010 and can be inspected alongside the HierCC clusters. Sequence type, cgMLST clusters at the HC0, HC900 (eBurst groups) and HC2000 (super-lineages in *Salmonella*) levels were also included in Supplementary Data 1.

Enterobase was searched on 21/07/2022 for additional *Salmonella* isolates that were part of three HC2000 clusters that harboured *S.* Concord isolates. This dataset can be accessed via https://enterobase.warwick.ac.uk/species/senterica/search_strains?query=workspace:79432.

A detailed *S.* Concord super-lineage A (HC_2000_750) SNP-based phylogeny was inferred from genome sequencing data of the 245 *S.* Concord isolates part of this super-lineage. In brief, snippy (https://github.com/tseemann/snippy) v4.6.0 was used to produce a core single nucleotide polymorphism (SNP) alignment of 20641 polymorphic SNP sites. A large proportion of these sites were flagged by Gubbins[74] v3.2.1 as recombinant. The remaining 7148 non-recombinant variant sites were extracted using snp-sites (https://github.com/sanger-pathogens/snp-sites) v2.5.1 and used to infer a phylogenetic tree using RAxML-NG[75] v0.9.0, employing the general time-reversible model with gamma-distributed rate heterogeneity and the Lewis ascertainment bias correction (--model GTR + G + ASC_LEWIS). The tree with the best maximum likelihood score out of 100 trees was combined with support values obtained from 1000 bootstrap replicates. For outgroup rooting of the *S.* Concord super-lineage A SNP tree, the *S. enterica* serovar Potsdam (*S.* Potsdam) isolate with accession SRR10962428 was selected as it was part of a small group of four *S.* Potsdam isolates within the HC2000_750 cgMLST tree that branched off earlier from all remaining HC2000_750 isolates which were *S.* Concord.

Core gene alignments were used to infer maximum likelihood phylogenies for *S.* Concord super-lineage B (*n* = 23) and C (*n* = 15). In brief, Prokka's gff files were used as input for Roary[76] v3.13.0 to generate a core gene alignment using PRANK[77] v.170427. Alignments were passed to RAxML-NG to infer a phylogeny under the GTR + G model. The tree with the best maximum likelihood score out of 100 trees was retained and midpoint rooted.

Ggtree[78] v2.2.4 was used to construct phylogenetic tree visualisations in R. Pairwise SNP distances for *S.* Concord super-lineage A were calculated using snp-dists (https://github.com/tseemann/snp-dists) v0.7.0 from the same SNP alignment used for phylogenetic inference, but without recombination removal. The resulting SNP matrix was visualised using ComplexHeatmap[79] v2.8.0 in R.

## Identification of lineages within super-lineage A

Fastbaps[80] v1.0.7 was used to detect clusters in the *S.* Concord super-lineage A (HC2000_750) phylogeny. Level 1 BAPS clusters were designated as lineages. BAPS cluster 1 was split into two lineages based on visual inspection and BAPS level 2 clustering. BAPS cluster 5 was paraphyletic and therefore split into 4 monophyletic lineages. All resulting lineages were monophyletic groups. The *S.* Potsdam isolate used to root the tree was excluded from lineage naming.

## Nanopore sequencing and structural differences in AMR regions

Ten isolates selected to represent multiple *S.* Concord lineages under super-lineage A and genomic AMR profiles, were sequenced using the MinIon sequencer (Oxford Nanopore Technologies, Oxford, UK). High molecular weight DNA was extracted from an overnight culture using the Nanobind CBB Big DNA Kit (Circulomics, Baltimore, USA) according to the manufacturer's protocol. At least 1 μg up to 1.5 μg of DNA was used as input for library prep with the Oxford Nanopore Technologies (ONT) Ligation Sequencing Kit (SQK-LSK109) combined with the Native Barcoding Kit (EXP-NBD104). Multiplexed samples were sequenced on two R9.4.1 flow cells for 66 h, following the ONT protocol for native barcoding of genomic DNA. Guppy (ONT) v4.4.0 was used for offline basecalling in high accuracy mode, demultiplexing and barcode trimming. Read length distributions and yields per barcode were inspected with PycoQC[81] v2.5.2. Taxonomic classification

with Kraken2 confirmed that all reads originated from the genus *Salmonella* and were not a contaminant. FASTQ files were filtered using filtlong (https://github.com/rrwick/Filtlong) v0.2.0, retaining 95% of the highest quality reads of at least 1000 base pairs (bp). FASTQ files for each isolate were submitted to the ENA and accessions are listed in Supplementary Data 1.

Candidate assemblies were generated using Canu[82] v2.1.1, Redbean[83] v2.5, Raven[84] v1.3.0 and Flye[85] v2.8.2, and used to build one consensus sequence for each replicon per sample with Trycycler[86] v0.4.1. The resulting consensus assemblies were polished with ONT reads using Medaka (https://github.com/nanoporetech/medaka) v1.2.2, utilising the r941_min_high_g360 model. Consequently, three rounds of Illumina read polishing were performed using Pilon[87] v1.23 with the minimum depth, minimum mapping quality and minimum base quality set to 10, 60 and 20 respectively. Nanopore reads were mapped back to the resulting references using minimap2 to assess the mean sequencing depth per contig as calculated with Mosdepth[88] v0.3.1. The mean depth of coverage for each contig per reference genome is reported in Supplementary Table 3 and ranged from 43 to 1881.

Genomes were annotated and queried for AMR genes and mutations as described under 'Annotation and AMR determinant detection'. Prokka annotation files were searched for insertion sequences. Mauve[89] v2.4.1 was used to align reference genomes and detect large chromosomal insertions or deletions which were visualised with the genoplotR[90] package v0.8.11 in R. To assess the organisation of AMR genes in detail, AMRfinder output was parsed in R and visualised using gggenes (https://github.com/wilkox/gggenes) v0.3.1.

### Antimicrobial susceptibility testing

Antimicrobial susceptibility testing was performed on isolates from the collections of DTU-Food, ITM and UKHSA for a total 56 isolates (see 'Results' for criteria of selection). The minimum inhibitory concentration (MIC) was determined using the Sensititre (Thermo Fisher Scientific) broth microdilution system for 15 antimicrobials. Isolates cultivated on BD Columbia Agar with 5% sheep blood were processed according to the manufacturer's protocol for Gram-negative bacteria. Two customised Sensititre panels (plate codes BELITG1 and BELITM2) were used containing antimicrobials formerly used and presently recommended to treat invasive *Salmonella* infections: ampicillin, chloramphenicol, trimethoprim/sulfamethoxazole, ceftriaxone, ciprofloxacin, and azithromycin. Several other antibiotics, were included, namely meropenem, gatifloxacin, tigecycline, colistin, temocillin, and combination agents ceftolozane/tazobactam, ceftazidime/avibactam, piperacillin/tazobactam and meropenem/vaborbactam. Resistance was interpreted according to CLSI M100Ed31E[91]. Azithromycin resistance was interpreted as a MIC higher than the epidemiological cut-off for (i)NTS, which was reported to be 16 mg/L. For temocillin and tigecycline, the epidemiological cut-offs of 16 mg/L and 0.5 mg/L reported by the EFSA[92] were used.

### Reporting summary
Further information on research design is available in the Nature Portfolio Reporting Summary linked to this article.

## Data availability
The sequencing reads generated using Illumina, Nanopore, and PacBio technologies can be accessed through ENA/SRA, and the corresponding accessions for each isolate can be found in Supplementary Data 1. Furthermore, assembled short-read data can be obtained from EnteroBase (https://enterobase.warwick.ac.uk/species/senterica/search_strains?query=workspace:79416). The data underlying Fig. 1 are accessible via Enterobase (https://enterobase.warwick.ac.uk/species/senterica/search_strains?query=workspace:79432), while the data underlying Fig. 2, including metadata, AMR genes, and replicon

genes, are available through Supplementary Data 1. Antimicrobial susceptibility testing data, which are the basis of Supplementary Figures 15 and 16, and Supplementary Table 2, can be obtained from Supplementary Data 2. The PlasmidFinder database (retrieved on March 1, 2020) is available at https://cge.food.dtu.dk/services/PlasmidFinder/, while the AMRFinder database (used database version: 2022-05-26.1) is accessible via https://www.ncbi.nlm.nih.gov/pathogens/antimicrobial-resistance/AMRFinder/. Enterobase was searched on 21/07/2022 for all isolates included in HC_2000, and the search result can be accessed at https://enterobase.warwick.ac.uk/species/senterica/search_strains?query=workspace:79432. Source data are provided with this paper.

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

## Acknowledgements

This work was financially supported by the Research Foundation - Flanders (FWO: SB Ph.D. fellowship 1S40018N to W. L. C.). The computational resources and services used in this work were provided by the HPC core facility CalcUA of the University of Antwerp, and the VSC (Flemish Supercomputer Center), funded by the Research Foundation - Flanders (FWO) and the Flemish Government. We acknowledge the support of BIOMINA (Biomedical Informatics Network Antwerpen). The work by S.V.P. and G.D. is funded in part by a grant from the Bill & Melinda Gates Foundation (OPP1151153). This research was funded by the National Institute for Health Research [Cambridge Biomedical Research Centre at the Cambridge University Hospitals NHS Foundation Trust]. K.M. and N.R.T. were supported by Wellcome Trust (grant number 206194). For the purpose of Open Access, the author has applied a CC-BY public copyright license to any Author Accepted Manuscript version

arising from this submission. M.A.C. is affiliated to the National Institute for Health Research Health Protection Research Unit (NIHR HPRU) in Genomics and Enabling Data at University of Warwick in partnership with the UK Health Security Agency (UKHSA), in collaboration with University or Cambridge and Oxford. M.A.C. is based at UKHSA and the views expressed are those of the authors and not necessarily those of the NHS, UKHSA, the NIHR or the Department of Health and Social Care. The laboratory of F.X.W. belongs to the "Integrative Biology of Emerging Infectious Diseases" Laboratory of Excellence funded by the French Government "Investissement d'Avenir" programme (grant no. ANR-10-LABX-62-IBEID). The French National Reference Center for *Escherichia coli*, *Shigella* and *Salmonella* is co-funded by Santé Publique France and the Institut Pasteur. Genome sequencing performed at the Earlham Institute as part of the 10KSG consortium was supported by the Global Challenges Research Fund data and resources grant (BBS/OS/GC/000009D), and the BBSRC National Capability in Genomics and Single Cell (BB/CCG1720/1) grant via members of the Genomics Pipelines Group. J.W. acknowledges the support of the Biotechnology and Biological Sciences Research Council through the Institute Strategic Programme Microbes in the Food Chain BB/R012504/1 and its constituent project BBS/E/F/000PR10349. We thank Ellen Corsmit, Tine Vermoesen, Véronique Guibert, Magali Ravel, and Estelle Serre for their assistance and/or advice concerning the cultivation of isolates and antimicrobial susceptibility testing, and Philippe Selhorst and Bart Cuypers for their advice on Nanopore sequencing.

## Author contributions

Study design and oversight: W.L.C., S.V.P., J.J., K.L., P.M, S.D. Isolate collection and metadata: F.X.W., R.S.H., G.B., J.W., M.A.C., S.N., P.-J.C., C.K., J.M-G., K.T.V., M.C., M.T., P.I.F., T.C., J.J. Wet lab: W.L.C., L.H., J.J., S.V.P., T.dB. Sequencing: K.C.M., N.T., F.X.W., M.P.G., B.P.S., G.D., J.W., W.L.C. Data analysis: W.L.C. Writing—original draft preparation: W.L.C., S.V.P., J.J., B.T., S.D., P.M., K.L. Writing—critical review: W.L.C., S.V.P., F.X.W., J.J., R.S.H., J.W., S.N., M.A.C., M.C., B.P.S., J.M-G., P.I.F., S.D., P.-J.C, B.T., L.H., W.W.Y.L. All authors have read and agreed to the published version of the manuscript.

## Competing interests

The authors declare no competing interests.

## Additional information

Wim L. Cuypers [1,2] ✉, Pieter Meysman [1], François-Xavier Weill [3], Rene S. Hendriksen [4], Getenet Beyene [5], John Wain [6,7], Satheesh Nair [8], Marie A. Chattaway [8], Blanca M. Perez-Sepulveda [9], Pieter-Jan Ceyssens [10], Tessa de Block [11], Winnie W. Y. Lee [8,12], Maria Pardos de la Gandara [3], Christian Kornschober [13], Jacob Moran-Gilad [14], Kees T. Veldman [15], Martin Cormican [16], Mia Torpdahl [17], Patricia I. Fields [18], Tomáš Černý [19], Liselotte Hardy [2], Bieke Tack [2,20], Kate C. Mellor [21,22], Nicholas Thomson [21,22], Gordon Dougan [23], Stijn Deborggraeve [24], Jan Jacobs [2,20], Kris Laukens [1] & Sandra Van Puyvelde [22,23,25] ✉

[1]Adrem Data Lab, Department of Computer Science, University of Antwerp, Antwerp, Belgium. [2]Unit of Tropical Bacteriology, Department of Clinical Sciences, Institute of Tropical Medicine, Antwerp, Belgium. [3]Institut Pasteur, Université Paris Cité, Unité des bactéries pathogènes entériques, F-75015 Paris, France. [4]Technical University of Denmark, National Food Institute (DTU-Food), Research Group of Global Capacity Building, Kgs., Lyngby, Denmark. [5]Department of Medical Laboratory Sciences, Faculty of Health Sciences, Jimma University, Jimma, Ethiopia. [6]Quadram Institute Bioscience, Norwich Research Park, Norwich, UK. [7]Norwich Medical School, University of East Anglia, Norwich, UK. [8]Gastrointestinal Bacterial Reference Unit, United Kingdom Health Security Agency, Colindale, London, UK. [9]Institute of Infection, Veterinary & Ecological Sciences, University of Liverpool, Liverpool, UK. [10]Division of Human Bacterial Diseases, Sciensano, Brussels, Belgium. [11]Clinical Reference Laboratory, Department of Clinical Sciences, Institute of Tropical Medicine, Antwerp, Belgium. [12]MRC Centre for Molecular Bacteriology and Infection, Imperial College London, London, UK. [13]Austrian Agency for Health and Food Safety (AGES), Institute for Medical Microbiology and Hygiene, 8010 Graz, Austria. [14]Department of Health Policy and Management, School of Public Health, Faculty of Health Sciences, Ben Gurion University of the Negev, Beer Sheva, Israel. [15]Department of Bacteriology, Host Pathogen Interaction & Diagnostics, Wageningen Bioveterinary Research, Lelystad, The Netherlands. [16]Antimicrobial Resistance and Microbial Ecology Group, School of Medicine, University of Galway, Galway, Ireland. [17]Department of Bacteriology, Mycology & Parasitology, Statens Serum Institut, 5 Artillerivej, DK-2300 Copenhagen S, Denmark. [18]Division of Foodborne, Waterborne and Environmental Diseases, Centers for Disease Control and Prevention, Atlanta, GA, USA. [19]National Reference Laboratory for salmonella, State Veterinary Institute Prague, Prague, Czech Republic. [20]Department of Microbiology, Immunology and Transplantation, KU Leuven, Leuven, Belgium. [21]London School of Hygiene and Tropical Medicine, Bloomsbury, London, UK. [22]Wellcome Trust Sanger Institute, Genome Campus, Hinxton, Cambridge, United Kingdom. [23]Cambridge Institute of Therapeutic Immunology & Infectious Disease (CITIID),Department of Medicine,  University of

Cambridge, Cambridge CB2 0SP, United Kingdom. [24]Department of Biomedical Sciences, Institute of Tropical Medicine, Antwerp, Belgium. [25]Laboratory of Medical Microbiology, Vaccine & Infectious Disease Institute, University of Antwerp, Antwerp, Belgium. ✉e-mail: wim.cuypers@uantwerpen.be; sandra.vanpuyvelde@uantwerpen.be

