## [Peer review file · Nature Communications]

REVIEWERS' COMMENTS

Reviewer #1 (Remarks to the Author):

This paper reports an in-depth genomic analysis of an understudied *Salmonella* serovar, *S. Concord*, that appears to be endemic in Ethiopia (an understudied LMIC). The findings demonstrate polyphyletic structure of serovar *S. Concord*, which calls for WGS-based surveillance to better understand its epidemiology. The phenotypic antimicrobial resistance data additionally strengthen this study by providing much needed clinical resistance data that could inform further clinical and pharmacokinetic studies aimed at collecting data to support treatment recommendations. This study also found a decrease in MDR, ESBL in 2014, which remains unexplained. As authors point out that this could be due to the isolate collection bias and the lack of data. Although there is evidence of recent foodborne transmission of *S. Concord*, this study does not comprehensively elucidate the reservoirs of resistant *S. Concord*.

Specific comments:

-L224: Could you please provide a justification for using `-carefull` and `-only` assembler options for genome assembly? Why was also error correction option not used, or alternatively, `-- isolate` option, which is recommended for bacterial isolate genome assembly?

-L460: Please provide more information about the *IncHI2A/IncA/C2* hybrid plasmid reconstruction. Were *IncHI2A/IncA/C2* markers detected on a single circular contig? How many plasmids in total were detected in isolate ERR9516323? Is it possible that a plasmid misassembly occurred, resulting in a hybrid artifact?

Reviewer #2 (Remarks to the Author):

The study presents a comprehensive and international survey of *S. Concord* genomes that fills the gaps in the genomic understanding of the population structure and AMR dissemination of this infrequently reported serotype. It provides convincing evidence that *S. Concord* lineages historically associated with Ethiopia have acquired high levels of AMR via plasmids. This finding echoes reports of other *Salmonella* strains circulating in sub-Saharan Africa that were likely driven to become MDR, XDR or PDR by the prevailing use of antibiotics to treat iNTS infections.

Methodologies and conclusions are solid and in line with what would be expected for this type of genomic epidemiology studies, especially when it comes to resolving AMR cassettes using long reads. The authors also discussed about the possibility of human reservoirs and human-to-human transmission of *S. Concord* in Ethiopia, which has been hypothesized for other “iNTS” strains in sub-Saharan Africa and warrants further investigations.

Specific comments:

L371-372: Delete “remaining” and change “had” to “that”?

L420-421: Change “were introduced first in the population” to “were first observed/detected in...” No analysis was done to estimate when these genes were introduced in the *S. Concord* population. MCRA dating of Ethiopian lineages would be helpful to show when these lineages merged in the context of the HIV pandemic and the emergence of other iNTS strains in SSA.

L518-519: This sentence needs to be edited for better clarity.

Reviewer #3 (Remarks to the Author):

Invasive Non-typhoidal salmonella cause disproportionately high morbidity and mortality in children from sub-Saharan Africa. The most studied serovar has been *S. Typhimurium* ST313. This paper describes an epidemiologic and genomics study on *S. Concord*, a serovar less studied and interestingly causing similar severe disease as *Typhimurium*. The team analysed *S. Concord* genomes comprising 284 isolates from 12 countries and spanning 78 years. They found that *S. Concord* is a diverse polyphyletic serovar spread across three *Salmonella* super-lineages. In addition, the data revealed that *S. Concord* lineages present in Ethiopia have developed a remarkably high level of Multi-drug resistance through horizontal gene transfer. This study is unique as strains of *S. Concord* lineages were rarely found outside of Ethiopia except through international travel, and even so, strains from USA foodborne outbreaks are usually fully susceptible to antibiotics and rarely invasive. The example of *S. Concord* merge, evolution and

spread illustrates how AMR can disseminate globally through human travel. Susceptible S. Concord strains appear to have disseminated recently through the food chain.

The study is well designed, and data have been thoroughly analysed to obtain phylogenetic relatedness trends over time. The conclusions are clearly drawn from data presented in the study and will make a major contribution to our current knowledge regarding this emerging and important Salmonella serovar.

Concerns not addressed:

1. What were the major genetic differences between the epidemic S. Concord lineages and the foodborne ones that were isolated from outbreaks in the USA?
2. What genetic differences would you attribute to invasive characteristics of the Ethiopian Concord Serovar compared to the foodborne strain types?

REVIEWERS' COMMENTS

Reviewer #1 (Remarks to the Author):

This paper reports an in-depth genomic analysis of an understudied *Salmonella* serovar, *S. Concord*, that appears to be endemic in Ethiopia (an understudied LMIC). The findings demonstrate polyphyletic structure of serovar *S. Concord*, which calls for WGS-based surveillance to better understand its epidemiology. The phenotypic antimicrobial resistance data additionally strengthen this study by providing much needed clinical resistance data that could inform further clinical and pharmacokinetic studies aimed at collecting data to support treatment recommendations. This study also found a decrease in MDR, ESBL in 2014, which remains unexplained. As authors point out that this could be due to the isolate collection bias and the lack of data. Although there is evidence of recent foodborne transmission of *S. Concord*, this study does not comprehensively elucidate the reservoirs of resistant *S. Concord*.

We are grateful for the time and effort the reviewer took to review our manuscript, and for the specific comments.

Specific comments:

- L224: Could you please provide a justification for using `--carefull` and `--only` assembler options for genome assembly? Why was also error correction option not used, or alternatively, `--isolate` option, which is recommended for bacterial isolate genome assembly?

In response to the reviewer's comment regarding our decision to use the `--careful` option instead of the `--isolate` option during the assembly of the *S. Concord* dataset, we would like to provide a more detailed explanation. The `--isolate` option is primarily designed for WGS data of bacterial isolates with a high depth of coverage (depth > 100, as detailed in the release notes of Spades v3.14.0 <https://github.com/ablab/spades/releases>), where its use can result in faster runtimes. However, it should be noted that certain isolates within the *S. Concord* dataset possessed a lower depth of coverage (40X), making the `--isolate` option less suitable. As a result, we chose to utilize the `--careful` option, which incorporates a polishing step to reduce the number of mismatches and short indels. It is important to note that the `--isolate` option disables this polishing step.

We do acknowledge the potential benefit of omitting the `--only-assembler` option in enhancing assembly quality through read error correction. Although read error correction may contribute minimally to improved assembly quality, or even deteriorate assembly metrics (as supported by literature resources such as reference Heydari et al. (2019)¹), we conducted an analysis to objectively evaluate whether enabling this option would enhance the assembly quality for the *S. Concord* dataset.

In brief, we ran spades with the `--careful` option enabled, but without the `--only-assembler` option, thereby enabling read error correction. Subsequently, we assessed several assembly quality metrics, indicating that, on average, enabling read error correction indeed slightly improves the assembly quality (see Table below).

Settings	New settings (<code>--careful</code>)	Original settings (<code>--careful --only- assembler</code>)
Mean number of contigs	68.34	74.48
Mean largest contig	846849.20	822829.40
Mean contig N50	331157.90	325431

Using the same parameters and software versions as described in the manuscript we used the new assemblies to identify AMR genes and replicon genes. The results were identical to the results obtained previously. We updated line 552 in the manuscript to reflect the new settings, and used the new assemblies to generate the data presented in Supplementary Figure 4 and Supplementary Table 2.

- L460: Please provide more information about the IncHI2A/IncA/C2 hybrid plasmid reconstruction. Were IncHI2A/IncA/C2 markers detected on a single circular contig? How many plasmids in total were detected in isolate ERR9516323? Is it possible that a plasmid misassembly occurred, resulting in a hybrid artifact?

We utilized a 'long read first' approach and performed multiple alignment and re-mapping steps to minimize assembly artifacts while reconstructing the IncHI2A/IncA/C2 hybrid plasmid. The optimal consensus assembly was identified after generating various candidate assemblies using different software approaches (Canu, Redbean, Raven, and Flye). We used the Tricycler² tool for this purpose in accordance with the publication of Wick et al. (2021)², as described in the methods section of the manuscript. Resultantly, we believe that our approach was thorough and considerate, and our choices resulted the most accurate and reliable results possible.

We detected exclusively the IncHI2A and IncA/C2 plasmids types in both the short-read and long-read assemblies of isolate ERR9516323. The presence of hybrid plasmid types in *S. Concord* was previously demonstrated using PCR by Fabré et al. (2009)³, highlighting the possibility that hybrid plasmids are present in *S. Concord*. However, we acknowledge that an assembly represents the best possible hypothesis of the genome structure, and any assembly approach can be subject to error. Therefore, we have updated lines 318 – 321 in the manuscript to reflect this limitation, while highlighting that the hybrid plasmid assembled in a circular contig:

“

A 415 kb circular contig of a complete L5 genome contained backbone elements originating from both IncHI2 and IncA/C2 plasmids present in L3 and L5 (Supplementary Figure 14). It is highly probable that this was an IncHI2A/IncA/C2 hybrid plasmid.

“

Reviewer #2 (Remarks to the Author):

The study presents a comprehensive and international survey of *S. Concord* genomes that fills the gaps in the genomic understanding of the population structure and AMR dissemination of this

infrequently reported serotype. It provides convincing evidence that *S. Concord* lineages historically associated with Ethiopia have acquired high levels of AMR via plasmids. This finding echoes reports of other *Salmonella* strains circulating in sub-Saharan Africa that were likely driven to become MDR, XDR or PDR by the prevailing use of antibiotics to treat iNTS infections.

Methodologies and conclusions are solid and in line with what would be expected for this type of genomic epidemiology studies, especially when it comes to resolving AMR cassettes using long reads. The authors also discussed about the possibility of human reservoirs and human-to-human transmission of *S. Concord* in Ethiopia, which has been hypothesized for other “iNTS” strains in sub-Saharan Africa and warrants further investigations.

We thank the reviewer for the positive feedback. Specific comments are addressed below.

Specific comments:

- L371-372: Delete “remaining” and change “had” to “that”?

We removed “remaining”. However, we kept “had” in this sentence, to ensure that it is unequivocally understood that L1 and L2 had no confirmed link to Ethiopia.

- L420-421: Change “were introduced first in the population” to “were first observed/detected in...” No analysis was done to estimate when these genes were introduced in the *S. Concord* population. MCRA dating of Ethiopian lineages would be helpful to show when these lineages merged in the context of the HIV pandemic and the emergence of other iNTS strains in SSA.

This sentence was changed to “were first detected in the population from 2003 onwards”. We agree with the reviewer that MCRA dating of Ethiopian lineages would be helpful. We aimed to perform a Bayesian analysis for this purpose, but due to the lack of a temporal signal in the data ($R^2 = 0.05$; verified using BactDating’s ‘roototip’ function in R and TempEst), we could not generate reliable estimates for MRCA dating.

- L518-519: This sentence needs to be edited for better clarity.

Lines 381 – 384 in the revised manuscript were edited for clarity:

The original sentence:

“

Highly resistant *S. Concord* was not confined to the orphanage environment in Ethiopia, but likely picked up at higher rates upon international adoptees coming from orphanages and thus introducing a sample bias.

“

was changed to:

“

Hence, highly resistant *S. Concord* identified in the study was not limited to the orphanage environment in Ethiopia, but instead may have been identified at higher rates by reference laboratories in Europe and the USA via adoptees originating from Ethiopian orphanages, which introduced a sampling bias.

“

Reviewer #3 (Remarks to the Author):

Invasive Non-typhoidal salmonella cause disproportionately high morbidity and mortality in children from sub-Saharan Africa. The most studied serovar has been *S. Typhimurium* ST313. This paper describes an epidemiologic and genomics study on *S. Concord*, a serovar less studied and interestingly causing similar severe disease as *Typhimurium*. The team analysed *S. Concord* genomes comprising 284 isolates from 12 countries and spanning 78 years. They found that *S. Concord* is a diverse polyphyletic serovar spread across three *Salmonella* super-lineages. In addition, the data revealed that *S. Concord* lineages present in Ethiopia have developed a remarkably high level of Multi-drug resistance through horizontal gene transfer. This study is unique as strains of *S. Concord* lineages were rarely found outside of Ethiopia except through international travel, and even so, strains from USA foodborne outbreaks are usually fully susceptible to antibiotics and rarely invasive. The example of *S. Concord* emergence, evolution and spread illustrates how AMR can disseminate globally through human travel. Susceptible *S. Concord* strains appear to have disseminated recently through the food chain.

The study is well designed, and data have been thoroughly analysed to obtain phylogenetic relatedness trends over time. The conclusions are clearly drawn from data presented in the study and will make a major contribution to our current knowledge regarding this emerging and important *Salmonella* serovar.

We want to express our gratitude to the reviewer for the valuable feedback provided. We addressed the raised concerns as outlined below, and believe these are valuable additions to the manuscript.

Concerns not addressed:

- What were the major genetic differences between the epidemic *S. Concord* lineages and the foodborne ones that were isolated from outbreaks in the USA?

The major genetic differences between foodborne lineages (lineages 6 and 7) and lineages from Ethiopia (lineage 3, 4, 5 and 8) in terms of the accessory genome, focussing on AMR genes and plasmid replicon genes were discussed in the manuscript. Foodborne isolates did not harbour AMR markers and plasmid replicon genes, while the epidemic *S. Concord* lineages from Ethiopia harboured large amounts of these markers (**Figure 2**). Something we had not discussed in the manuscript is how the SNP diversity within these lineages relates to other lineages. Therefore, we added **Supplementary Figure 6**, and edited line 254 in the manuscript to highlight that there was little genetic variation compared to other lineages.

- What genetic differences would you attribute to invasive characteristics of the Ethiopian Concord Serovar compared to the foodborne strain types?

Beyene et al. (2011)⁴ reported a high invasiveness of Ethiopian *S. Concord* in infants from Ethiopia. It is important to acknowledge that the observed invasiveness does not necessarily imply the existence of intrinsic invasiveness in *S. Concord*, resulting from certain genomic adaptations. It is also possible that certain patient factors (age, weakened immune system, ...) contributed to the high invasiveness observed by Beyene et al. (2011)⁴.

An analysis linking genetic variations to the presence or absence of isolates in the patient's bloodstream could not be conducted, as the availability of blood cultures in addition to stool cultures was unknown for numerous isolates. Therefore, to gain a more comprehensive understanding of the potential intrinsic invasiveness of *S. Concord*, we employed the machine learning method developed by Wheeler et al. (2018)⁵ to screen for lineages with a higher invasiveness index. At the core of this method is the detection of impactful mutations in specific genes that have been previously linked to increased invasiveness. The method has been previously used to identify *Salmonella* lineages with a potentially higher invasiveness^{6,7}.

Our findings suggest that Lineage 4, which was linked to Ethiopia, harboured more impactful mutations in genes previously linked to an increased invasive potential of isolates. **Supplementary Figure 4** was added showing these results, and source data was made available. A further detailed assessment of all of these genes showed that 23 genes showed gene presence-absence variations in multiple lineage 4 isolates (**Supplementary Table 2**). Lines 240-244 were added to the results section of the revised manuscript detailing these findings.

References:

1. Heydari, M., Miclotte, G., Van de Peer, Y. & Fostier, J. Illumina error correction near highly repetitive DNA regions improves de novo genome assembly. *BMC Bioinformatics* **20**, 298 (2019).
2. Wick, R. R. *et al.* Tricycler: consensus long-read assemblies for bacterial genomes. *Genome Biol.* **22**, 266 (2021).
3. Fabre, L. *et al.* Chromosomal integration of the extended-spectrum β -lactamase gene bla CTX-M-15 in *Salmonella enterica* serotype Concord isolates from internationally adopted children. *Antimicrob. Agents Chemother.* **53**, 1808–1816 (2009).
4. Beyene, G. *et al.* Multidrug resistant *Salmonella Concord* is a major cause of salmonellosis in children in Ethiopia. *J. Infect. Dev. Ctries.* **5**, 023–033 (2011).
5. Wheeler, N. E., Gardner, P. P. & Barquist, L. Machine learning identifies signatures of host adaptation in the bacterial pathogen *Salmonella enterica*. *PLoS Genet.* **14**, e1007333 (2018).

6. Van Puyvelde, S. *et al.* An African Salmonella Typhimurium ST313 sublineage with extensive drug-resistance and signatures of host adaptation. *Nat. Commun.* **10**, 4280 (2019).
7. Bawn, M. *et al.* Evolution of Salmonella enterica serotype Typhimurium driven by anthropogenic selection and niche adaptation. *PLoS Genet.* **16**, e1008850 (2020).